# The Effect of Nb-Content on the Microstructures and Corrosion Properties of CrFeCoNiNb_x_ High-Entropy Alloys

**DOI:** 10.3390/ma12223716

**Published:** 2019-11-11

**Authors:** Chun-Huei Tsau, Chen-Yu Yeh, Meng-Chi Tsai

**Affiliations:** Institute of Nanomaterials, Chinese Culture University, Taipei 111, Taiwan; qazwsx61515@gmail.com (C.-Y.Y.); asd99586@yahoo.com.tw (M.-C.T.)

**Keywords:** CrFeCoNiNb_x_ alloys, microstructure, hardness, corrosion property

## Abstract

This work studied the effect of niobium-content on the microstructures, hardness, and corrosion properties of CrFeCoNiNb_x_ alloys. Results indicated that the microstructures of these alloys changed from granular structures (CrFeCoNi alloy) to the hypereutectic structures (CrFeCoNiNb_0.2_ and CrFeCoNi_0.4_ alloys), and then to the hypoeutectic microstructures (CrFeCoNiNb_0.6_ and CrFeCoNi alloys). The lattice constants of the major two phases in these alloys, fcc and Laves phases (hcp), increased with the increasing of Nb-content because of solid-solution strengthening. The hardness of CrFeCoNiNb_x_ alloys also had the same tendency. Adding niobium would slightly lessen the corrosion resistance of CrFeCoNiNb_x_ alloys in 1 M deaerated sulfuric acid and 1 M deaerated sodium chloride solutions, but the CrFeCoNiNb_x_ alloys still had better corrosion resistance in comparison with commercial 304 stainless steel. In these dual-phased CrFeCoNiNb_x_ alloys, the fcc phase was more severely corroded than the Laves phase after polarization tests in these two solutions.

## 1. Introduction

The high-entropy alloy concept has been announced for many years [1], and many important results have been published recently [2,3,4]. This concept creates new alloys from many elements with different atomic radiuses and crystal structures. The final structures of the high-entropy alloys are thus not dominated by any one of the elements because the molar fraction of each element is not high enough. The high-entropy alloy concept has four important core effects, which are: High entropy, lattice distortion, sluggish diffusion, and cocktail effects [4,5]. The high-entropy alloy concept is thus used to develop new alloys with special properties because of these four core effects. Corrosion resistance is also an important property for structural application of high-entropy alloys.

Many studies of the corrosion behaviors of high-entropy alloys have been published. Al_x_CrFe_1.5_MnNi_0.5_ alloy possesses good workability and high-temperature oxidation resistance [6]. The pitting resistance of Al_0.3_CrFe_1.5_MnNi_0.5_ alloy can be enhanced after anodic treatment [7] because a stable passive film can be formed on the surface. The corrosion resistance of Fe_73.5_Cu_1_Nb_x_Si_13.5_B_10_ alloy can be improved after adding a suitable amount of Nb [8]. Moreover, FeCoNiCrCu*_x_* alloy possesses a good corrosion resistance in a 3.5% sodium chloride solution [9]. Similar results are also observed in Al_7.5_Cr_22.5_Fe_35_Mn_20_Ni_15_ and Al_0.5_CoCrFeNi alloys [10,11]. These literatures proved that the high-entropy alloy concept is a good method to improve the corrosion resistance of structural materials.

Referring to our previous studies, CrFeCoNi alloy has an fcc granular structure, and some hcp Cr-rich precipitates are in the matrix [12]; it has a good corrosion resistance in 1 M deaerated sulfuric acid and 1 M deaerated sodium chloride solutions. However, the CrFeCoNi alloy is very soft, which limits its application. The hardness of CrFeCoNi alloy could be enhanced after the adding of Nb [13]. The corrosion resistance of CrFeCoNiNb alloy slightly decreases in comparison with that of CrFeCoNi alloy, but it is still better than that of commercial 304 stainless steel. The microstructures and mechanical properties of CrFeCoNiNb_x_ alloys are also investigated because of their high performance for commercial applications [14,15,16]. This study investigates the microstructures, hardness, and corrosion behaviors of CrFeCoNiNb_x_ (x = 0–1) alloys.

## 2. Experimental Section

The experimental CrFeCoNiNb_x_ alloys were prepared by an arc melter in an Ar atmosphere with a partial pressure of 200 torr, and the purities of Cr, Fe, Co, Ni, and Nb were all higher than 99.9%. The nominal compositions of CrFeCoNiNb_x_ alloys are listed in Table 1. The microstructures of CrFeCoNiNb_x_ alloys for observation were metallographically prepared and chemically etched in aqua regia (25% HNO_3_ and 75% HCl in volume fraction). The microstructures and chemical compositions of the alloys were done using a field emission scanning electron microscope with an energy dispersive spectrometer (SEM/EDS, JEOL JSM-6335, JEOL Ltd., Tokyo, Japan) operated at 15 kV. The phases in the alloys were identified by an X-ray diffractometer (XRD, Rigaku ME510-FM2, Rigaku Ltd., Tokyo, Japan) with Cu–Kα (with a wavelength of 1.541 Å) radiation operated at 30 kV. The microstructures and corresponding selection area diffraction patterns (SAD) of the alloys were observed by a transmission electron microscope (TEM, JEOL JEM-2000EX II, JEOL Ltd., Tokyo, Japan) operated at 200 kV. The TEM specimens were electrochemically prepared by a model 110 digital Fischione twin-jet electropolisher at a potential of 30 V (Fischione Instruments Co., Pittsburgh, PA, USA), and the solution was 10 vol.% perchloric acid and 90 vol.% methanol. The hardness of the alloys was measured using a Matsuzawa Seiki MV1 Vicker’s hardness tester (Matsuzawa Co., Akita, Japan) under a load of 294 N (30 kgf), and more than five points were averaged for each alloy.

Polarization curves of the alloys were tested in a potentiostat/galvanostat with a three-electrode system (Autolab PGSTAT302N, Metrohm Autolab B.V., Utrecht, the Netherlands). The exposed surface of each as-cast CrFeCoNiNb_x_ alloy for polarization testing was fixed at 0.1964 cm^2^ (with a diameter of 0.5 cm). The reference electrode was a saturated silver chloride electrode (Ag/AgCl), and the counter electrode was a Pt wire. The potential of a saturated silver chloride electrode (SSE, V_SSE_) is 222 mV higher than that of the standard hydrogen electrode (SHE, V_SHE_) at 25 °C [17]. The specimens were all mechanically polished using 1200 SiC grit paper. Test solutions were prepared from reagent-grade sulfuric acid (H_2_SO_4_) and sodium chloride (NaCl) that were dissolved in distilled water, and the concentrations of these two solutions were fixed at 1 M. To eliminate any effect of dissolved oxygen, the solutions were deaerated by bubbling nitrogen gas with a flow rate of 10 sccm/min through them before and during the polarization experiments. The immersion time before the experiment was fixed at 900 s for stabilizing the open circuit potential (*E*_OC_). The scanning rate was fixed at 1 mV/s, and the scanning range was from *E*_OC_ −1 V to *E*_OC_ +2 V in a single pass. The polarization data were also compared with those of commercial 304 stainless steel (304SS), and the composition of 304SS was 71.61% Fe, 18.11% Cr, 8.24% Ni, 1.12% Mn, 0.75% Si, 0.05% Co, 0.02% Mo, 0.05% C, 0.03% P, and 0.02% S by weight.

## 3. Results and Discussion

The as-cast SEM microstructures of CrFeCoNiNb_x_ alloys are shown in Figure 1. Figure 1a shows the microstructure of as-cast CrFeCoNi alloy. The CrFeCoNi alloy had a granular microstructure with an average grain size of several hundred micrometers; some hcp-structured Cr-rich precipitates in the fcc matrix were reported in our previous study [12]. However, the microstructures of as-cast CrFeCoNiNb_x_ alloys became a dendritic microstructure after adding niobium. The dendrites were a single phase, and the interdendrites showed a eutectic structure. The dendrites of these alloys had different morphologies after etching. The dendrites of CrFeCoNiNb_0.2_ and CrFeCoNiNb_0.4_ alloys had significant imprints after etching, as shown in Figure 1b,c, but those of CrFeCoNiNb_0.6_ and CrFeCoNiNb alloys exhibited almost smooth surfaces after etching, as shown in Figure 1d,e. These differences were caused by the different phases of the dendrites in these alloys. The dendrites of CrFeCoNiNb_0.2_ and CrFeCoNiNb_0.4_ alloys were an fcc phase, and the dendrites of CrFeCoNiNb_0.6_ and CrFeCoNiNb alloys were a Laves phase (hcp structure). The interdendrites of CrFeCoNiNb_x_ alloys had two major phases which were fcc and hcp phases (Laves phase). These were identified and described below.

Figure 2 shows the XRD patterns of CrFeCoNiNb_x_ alloys. There are two phases which existed in the CrFeCoNiNb_x_ alloys, which are fcc and Laves phases. The intensities of Laves phases in the CrFeCoNiNb_x_ alloys increase with increasing Nb-content. Table 2 lists the lattice constants of fcc and Laves phases analyzed from the XRD patterns. The lattice constants of fcc and Laves phases increased with increasing Nb-content in the CrFeCoNiNb_x_ alloys. This result was caused by the larger atomic radius of niobium. The atomic radiuses of Cr, Fe, Co, Ni, and Nb are 0.128, 0.124, 0.125, 0.125, and 0.143 nm, respectively [18]. The radiuses of Cr, Fe, Co, and Ni are very close, but niobium has a much larger atomic radius. Therefore, the lattice constants of fcc and Laves phases increased when the solid-solution amount of Nb-content increased in these alloys.

Figure 3 shows the TEM bright field (BF) images to examine the phases of the CrFrCoNiNb_x_ alloys, and the inserts are their corresponding select area diffraction patterns (SADs). Figure 3a is an image of the dendrite of CrFrCoNiNb_0.4_ alloy. The corresponding SAD proves that the dendrite is an fcc phase. Figure 3b is an image of the interdendrite of CrFrCoNiNb_0.4_ alloy, and the corresponding SAD was taken from the Laves phase in the interdendrite. Figure 3c is an image of the dendrite of CrFrCoNiNb_0.6_ alloy. The corresponding SAD proves that the dendrite is a Laves phase (hcp structure). Figure 3d shows the image of the interdendrite of CrFrCoNiNb_0.6_ alloy, and the corresponding SAD was taken from the fcc phase in the interdendrite of CrFrCoNiNb_0.6_ alloy. Therefore, Figure 3 indicates that the CrFeCoNiNb_0.4_ alloy is a hypoeutectic alloy and that the dendrites of this alloy are an fcc phase; the CrFeCoNiNb_0.6_ alloy is a hypereutectic alloy, and the dendrites of the alloy are the Laves phase. In all of these alloys, the interdendrites are a dual-phased (fcc and Laves phase) eutectic structure. Chanda and Das [14] also pointed out that the Nb-content of eutectic CrFeCoNiNb_x_ alloy should be located between 0.5 and 0.55.

Table 3 lists the chemical compositions of the CrFeCoNiNb_x_ alloys which were analyzed by SEM/EDS. The fcc phase had higher Cr-, Fe-, Co-, and Ni-contents, but less Nb-content than the average compositions of the alloys. On the contrary, the Laves phase of CrFeCoNiNb_x_ alloys had higher Nb-content and less Cr-, Fe-, Co-, and Ni-contents than the average.

Figure 4 plots the overall hardness of CrFeCoNiNb_x_ alloys as a function of x-ratio. The hardness of CrFeCoNiNb_x_ alloys almost linearly increases with increasing Nb-content. The two phases, fcc and Laves phases, in these CrFeCoNiNb_x_ alloys were two solid-solution phases, because no superlattice peak was observed from the XRD patterns and no superlattice spot was found from the TEM SADs of the CrFeCoNiNb_x_ alloys. Therefore, the effect of solid-solution strengthening was caused by the larger-radius element Nb in the fcc and Laves phases.

The polarization curves of CrFeCoNiNb_x_ alloys and 304SS in 1 M deaerated H_2_SO_4_ solution are shown in Figure 5, and Table 4 lists the polarization data of the alloys. The values of free corrosion potential (*E*_corr_) of these alloys are close. Increasing the Nb-content of CrFeCoNiNb_x_ alloys resulted in slightly increasing the *E*_corr_ of the alloys. The free corrosion current density (*i*_corr_) varied randomly in these alloys, because the microstructures and the fractions of the phases in the alloys changed significantly. The CrFeCoNiNb_0.6_ alloy had the lowest *i*_corr_ among these alloys. In addition, increasing the Nb-content of CrFeCoNiNb_x_ alloys resulted in decreasing the anodic peaks of these alloys. The 304 stainless steel had the largest anodic peak among these alloys, and the CrFeCoNiNb alloy had the smallest anodic peak. The largest current density (*i*_crit_) and the potential (*E*_pp_) of the anodic peak of each alloy are listed in Table 4. Therefore, the CrFeCoNiNb alloy entered the passivation region more easily than the other alloy in this solution. Moreover, adding Nb could stabilize the passivation region of CrFeCoNiNb_x_ alloys. The passivation regions of these alloys broke down at potentials of about 1.2 V_SHE_ because of an oxygen evolution reaction [19]. All of the passivation regions of CrFeCoNiNb_x_ alloys were broader than that of 304SS.

The morphologies of CrFeCoNiNb_x_ alloys after polarization tests in 1 M deaerated H_2_SO_4_ solution at 30 °C are shown in Figure 6. The morphology of CrFeCoNi alloy indicated that it was uniformly and slightly corroded after the polarization test, as shown in Figure 6a. Figure 6b shows the micrograph of CrFeCoNiNb_0.2_ alloy after the test. The fcc-dendrites of CrFeCoNiNb_0.2_ alloy were concave, which meant that the dendrites were more severely corroded than the interdendrites. This implied that the fcc phase was more active than the hcp phase; the fcc phase thus preferred to be corroded more than the Laves phase. Figure 6c clearly indicates this phenomenon: The fcc dendrites of CrFeCoNiNb_0.4_ alloy were more severely corroded and almost vanished. However, the dendrites of CrFeCoNiNb_0.6_ alloy changed to the Laves phase, and the dendrites almost kept their original shapes; the matrix of interdendrites (fcc phase) was corroded after the polarization test, as shown in Figure 6d. The CrFeCoNiNb alloy also had a similar result, shown in Figure 6e.

The polarization curves of CrFeCoNiNb_x_ alloys and 304SS in 1 M deaerated NaCl solution are shown in Figure 7, and Table 5 lists the polarization data of the alloys. The values of *E*_corr_ of CrFeCoNiNb_x_ alloys were very close; they are nobler than that of 304SS, but more active than that of CrFeCoNi alloy. The values of *i*_corr_ of CrFeCoNiNb_x_ alloys were all higher than that of CrFeCoNi alloy. The *i*_corr_ of CrFeCoNiNb_0.6_ alloy had the highest value. The CrFeCoNiNb_0.2_ and CrFeCoNiNb alloys had smaller *i*_corr_, and their values were almost the same as that of 304SS. The primary anodic peak of CrFeCoNi alloy was significant, but increasing the Nb-content of CrFeCoNiNb_x_ alloys would diminish the anodic peaks of these alloys. Therefore, the CrFeCoNiNb alloy had no anodic peak. Moreover, adding Nb could expand the passivation regions of CrFeCoNiNb_x_ alloys.

The morphologies of CrFeCoNiNb_x_ alloys after polarization tests in 1 M deaerated NaCl solution at 30 °C are shown in Figure 8. The micrograph of CrFeCoNi alloy after the polarization test indicated that it was only slightly corroded, as shown in Figure 8a. However, the fcc dendrites of CrFeCoNiNb_0.2_ and CrFeCoNiNb_0.4_ alloys were severely corroded after the tests, as shown in Figure 8b,c. The dendrites (Laves phase) of CrFeCoNiNb_0.6_ and CrFeCoNiNb alloys almost kept their original shapes, but the matrices of the interdendrites (fcc phase) were corroded after the tests, shown in Figure 8d,e. Therefore, the fcc phase was more active than the Laves phase in the CrFeCoNiNb_x_ alloys, and the fcc phase thus preferred to corrode more than Laves phase in 1 M deaerated H_2_SO_4_ and 1 M deaerated NaCl solutions.

## 4. Conclusions

The CrFeCoNi alloy had an fcc granular structure. The microstructures became dual-phased dendritic microstructures after adding niobium. The CrFeCoNiNb_0.2_ and CrFeCoNiNb_0.4_ alloys were hypereutectic alloys; their dendrites were fcc phases, and the interdendrites were eutectic structures of fcc and Laves phases (hcp). The CrFeCoNiNb_0.6_ and CrFeCoNiNb alloys were hypoeutectic alloys; their dendrites were a Laves phase, and the interdendrites were still eutectic structures of fcc and Laves phases.The lattice constants of fcc and Laves phases in these CrFeCoNiNb_x_ alloys increased with increasing Nb-content due to solid-solution strengthening. Increasing Nb-content also resulted in increasing the hardness of CrFeCoNiNb_x_ alloys. The hardness increased from HV144 of CrFeCoNi alloy to HV652 of CrFeCoNiNb alloy.The corrosion resistance of CrFeCoNiNb_x_ alloys slightly decreased after adding niobium because of their dual-phased dendritic microstructures. In addition, adding niobium into CrFeCoNiNb_x_ alloys could stabilize and expand the passivation regions of these alloys in these two solutions. The fcc phase of each CrFeCoNiNb_x_ alloy was more severely corroded than the Laves phase after polarization tests in 1 M deaerated H_2_SO_4_ and 1 M deaerated NaCl solutions.The CrFeCoNiNb_x_ alloys possessed good corrosion resistance, and their hardness changed with Nb-content. This work thus provides a method to design a CrFeCoNiNb_x_ alloy with suitable hardness and good corrosion resistance for different commercial applications.

## Figures and Tables

**Figure 1 materials-12-03716-f001:**
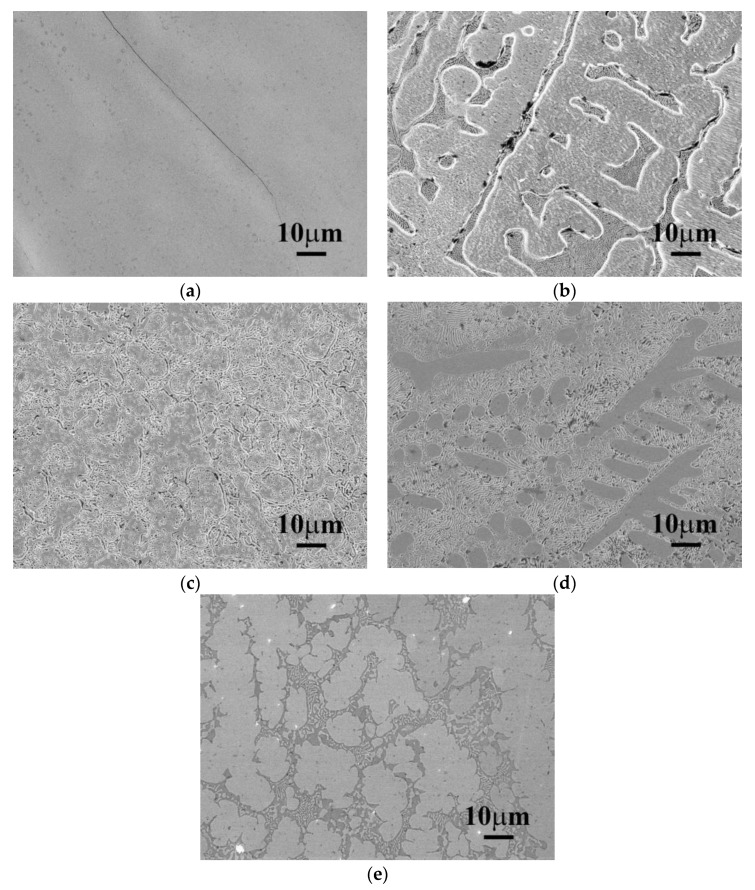
The SEM microstructures of as-cast CrFeCoNiNb_x_ alloys, (**a**) CrFeCoNi; (**b**) CrFeCoNiNb_0.2_; (**c**) CrFeCoNiNb_0.4_; (**d**) CrFeCoNiNb_0.6_; and (**e**) CrFeCoNiNb alloys.

**Figure 2 materials-12-03716-f002:**
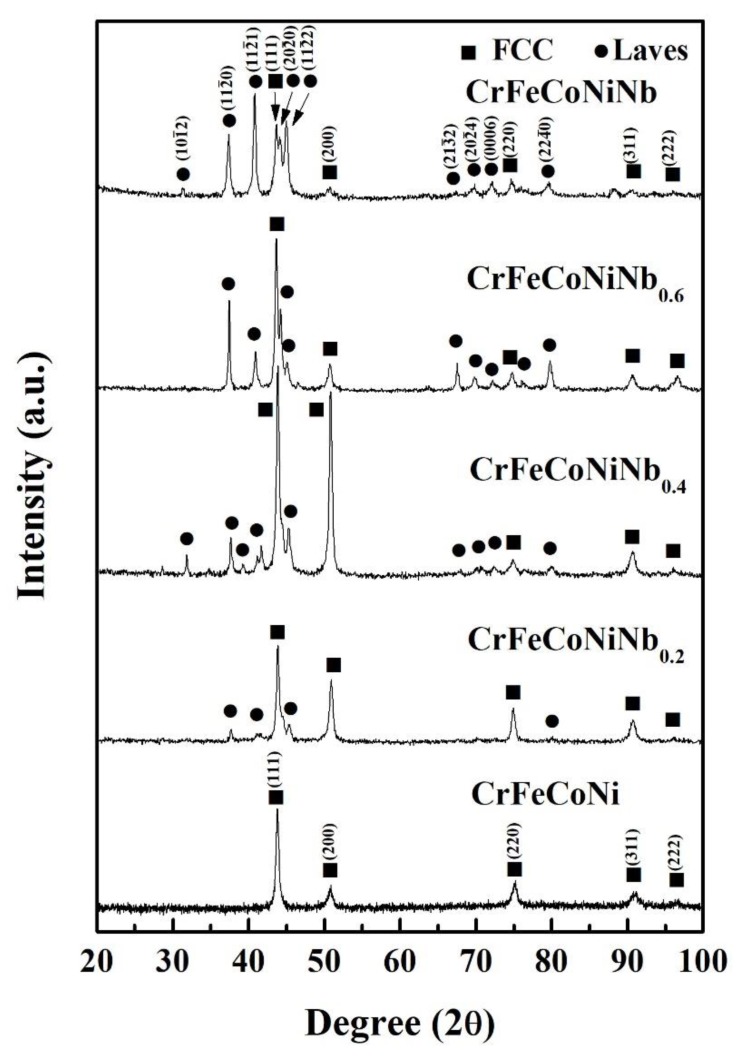
XRD patterns of CrFeCoNiNb_x_ alloys.

**Figure 3 materials-12-03716-f003:**
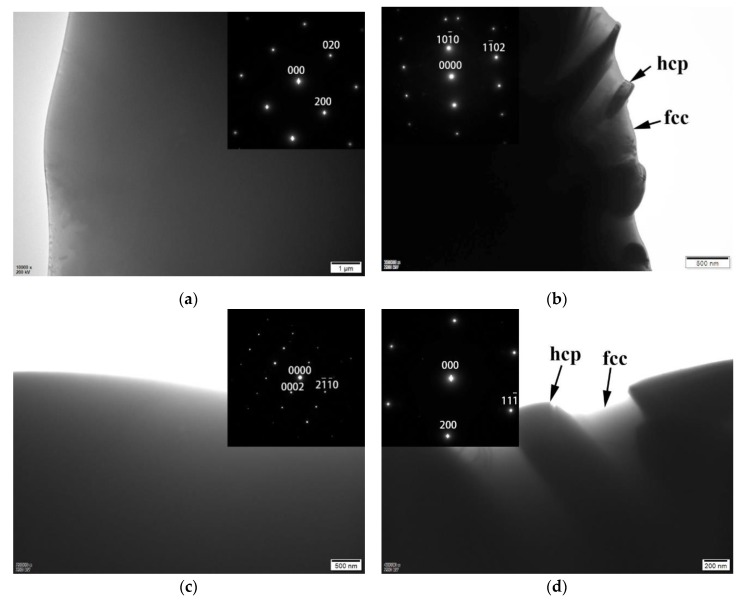
TEM bright field (BF) images and the corresponding selection area diffraction patterns (SAD) of CrFeCoNiNb_x_ alloys: (**a**) the dendrite of CrFeCoNiNb_0.4_ alloy, and the corresponding SAD taken from [001] of fcc phase; (**b**) the Laves phase in the interdendrite of CrFeCoNiNb_0.4_ alloy, and the corresponding SAD taken from [$\bar{2}$4$\bar{2}$3] of Laves phase; (**c**) the Laves phase in the dendrite of CrFeCoNiNb_0.6_ alloy, and the corresponding SAD taken from [01$\bar{1}$0] of Laves phase; and (**d**) the fcc phase in the interdendrite of CrFeCoNiNb_0.6_ alloy, and the corresponding SAD taken from [011] of fcc phase.

**Figure 4 materials-12-03716-f004:**
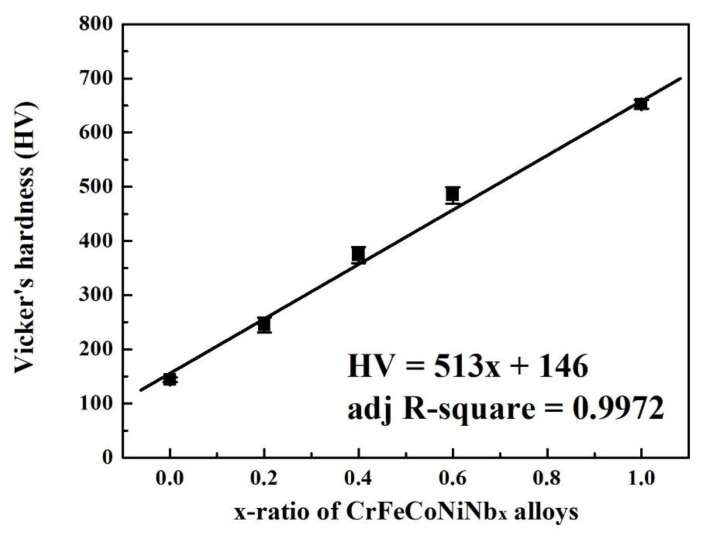
Plot of the overall hardness of CrFeCoNiNb_x_ alloys as a function of x-ratio.

**Figure 5 materials-12-03716-f005:**
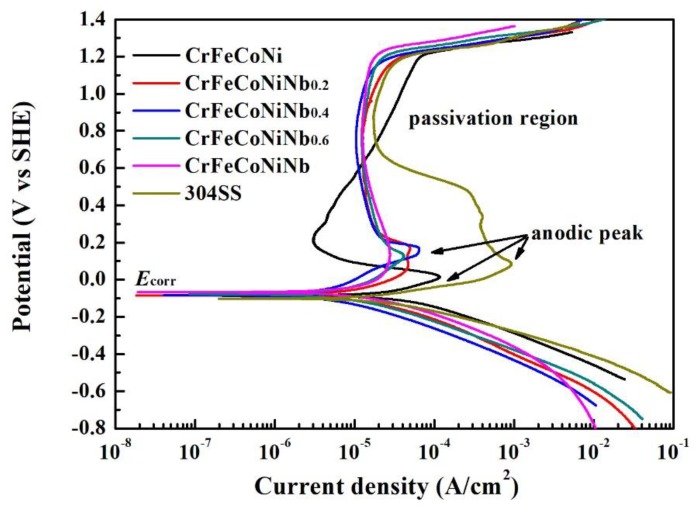
Polarization curves of CrFeCoNiNb_x_ alloys and 304SS in 1 M deaerated H_2_SO_4_ solution at 30 °C.

**Figure 6 materials-12-03716-f006:**
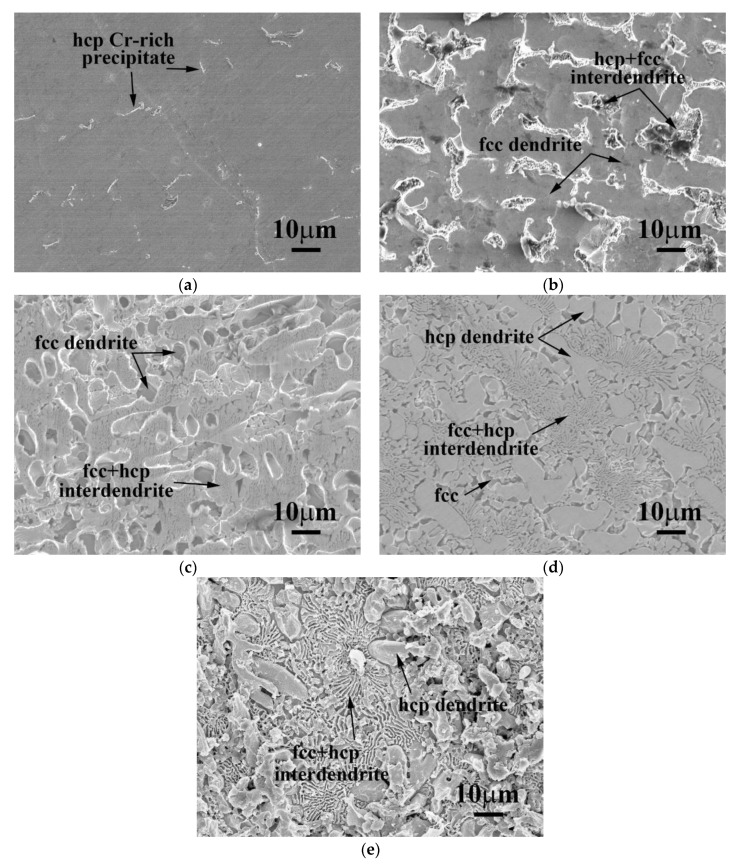
SEM micrographs of the alloys after polarization test in 1 M deaerated H_2_SO_4_ solution at 30 °C (**a**) CrFeCoNi; (**b**) CrFeCoNiNb_0.2_; (**c**) CrFeCoNiNb_0.4_; (**d**) CrFeCoNiNb_0.6_; and (**e**) CrFeCoNiNb alloys.

**Figure 7 materials-12-03716-f007:**
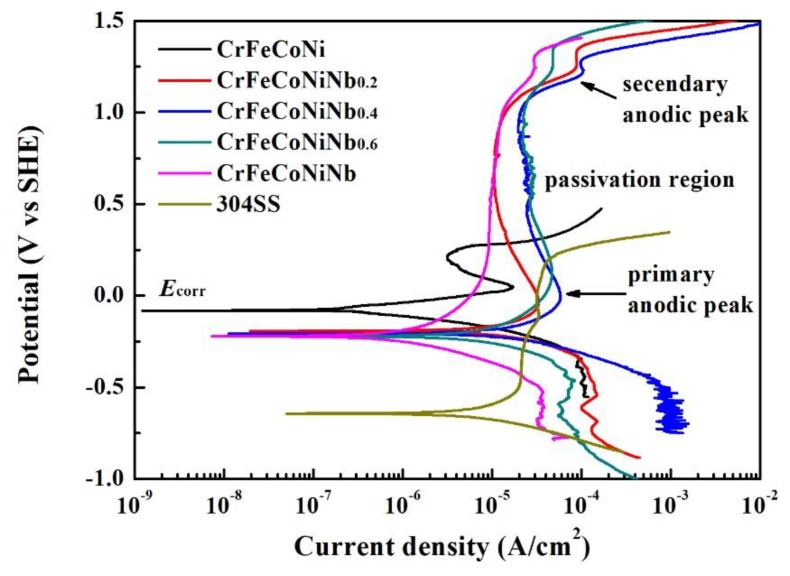
Polarization curves of CrFeCoNiNb_x_ alloys and 304SS in 1 M deaerated NaCl solution at 30 °C.

**Figure 8 materials-12-03716-f008:**
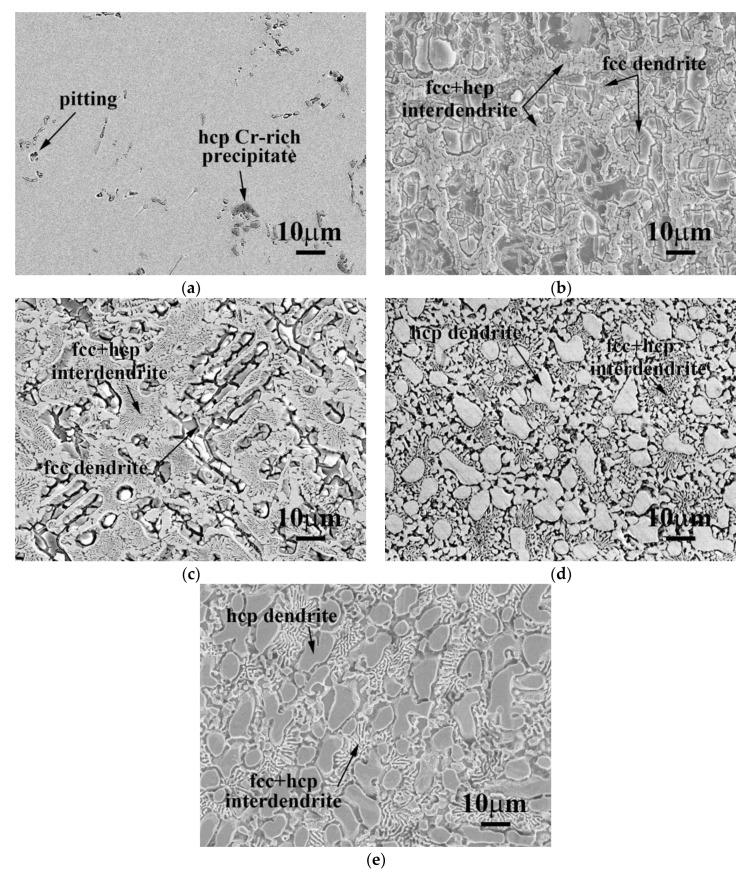
SEM micrographs of the alloys after polarization tests in 1 M deaerated NaCl solution at 30 °C (**a**) CrFeCoNi; (**b**) CrFeCoNiNb_0.2_; (**c**) CrFeCoNiNb_0.4_; (**d**) CrFeCoNiNb_0.6_; and (**e**) CrFeCoNiNb alloys.

**Table 1 materials-12-03716-t001:** Nominal compositions of CrFeCoNiNb_x_ alloys.

Alloys (at.%)	Compositions (wt.%)				
	Cr	Fe	Co	Ni	Nb
CrFeCoNi	23.06	24.77	26.13	26.04	N/A
CrFeCoNiNb_0.2_	21.31	22.88	24.14	24.05	7.62
CrFeCoNiNb_0.4_	19.80	21.26	22.44	22.35	14.15
CrFeCoNiNb_0.6_	18.49	19.86	20.95	20.88	19.82
CrFeCoNiNb	16.33	17.54	18.51	18.44	29.18

**Table 2 materials-12-03716-t002:** Lattice constants of fcc and Laves phases in as-cast CrFeCoNiNb_x_ alloys.

Alloys	fcc (Å)	Laves Phase (hcp) (Å)
CrFeCoNi	3.577	N/A
CrFeCoNiNb_0.2_	3.578	*a* = 4.773; *c* = 7.818
CrFeCoNiNb_0.4_	3.581	*a* = 4.773; *c* = 7.831
CrFeCoNiNb_0.6_	3.590	*a* = 4.798; *c* = 7.841
CrFeCoNiNb	3.590	*a* = 4.802; *c* = 7.848

**Table 3 materials-12-03716-t003:** Chemical compositions of the CrFeCoNiNb_x_ alloys analyzed by SEM/EDS.

Alloys	Compositions (Atomic Percent)				
	Cr	Fe	Co	Ni	Nb
CrFeCoNi					
Overall	23.1	24.8	26.1	26.0	N/A
Cr-rich precipitate	53.8	16.1	15.9	14.2	N/A
CrFeCoNiNb_0.2_					
Overall	23.4	23.5	23.3	22.3	7.4
fcc	24.2	26.5	24.3	23.2	1.8
hcp	19.4	18.2	23.1	21.2	18.1
CrFeCoNiNb_0.4_					
Overall	22.7	21.6	22.3	21.6	11.8
fcc	25.2	24.3	22.2	24.9	3.5
hcp	21.3	19.3	19.8	19.1	20.5
CrFeCoNiNb_0.6_					
Overall	21.4	20.6	21.5	19.3	17.2
fcc	25.1	24.4	21.8	23.0	5.7
hcp	16.5	18.4	22.3	15.9	26.9
CrFeCoNiNb					
Overall	19.8	19.8	19.0	19.4	22.0
fcc	26.7	24.3	18.4	25.5	5.1
hcp	17.1	18.1	19.7	14.8	30.3
precipitate	16.7	16.7	19.1	15.7	31.8

**Table 4 materials-12-03716-t004:** Polarization data of the alloys in 1 M deaerated H_2_SO_4_ solution at 30 °C.

Alloys	*E*_corr_ (V_SHE_)	*i*_corr_ (A/cm^2^)	*E*_pp_ (V_SHE_)	*i*_crit_ (A/cm^2^)
CrFeCoNi	−0.086	35.1	0.014	120
CrFeCoNiNb_0.2_	−0.084	15.8	0.082	47.1
CrFeCoNiNb_0.4_	−0.082	52.1	0.183	64.4
CrFeCoNiNb_0.6_	−0.078	10.1	0.122	41.9
CrFeCoNiNb	−0.068	22.3	0.132	28.0
304SS	−0.101	30.0	0.082	930

**Table 5 materials-12-03716-t005:** Polarization data of the alloys in 1 M deaerated NaCl solution at 30 °C.

Alloys	*E*_corr_ (V_SHE_)	*i*_corr_ (μA/cm^2^)	*E*_pp_ (V_SHE_)	*i*_crit_ (μA/cm^2^)
CrFeCoNi	−0.081	0.28	0.049	17.3
CrFeCoNiNb_0.2_	−0.192	12.3	−0.086	33.2
CrFeCoNiNb_0.4_	−0.207	21.1	−0.006	58.1
CrFeCoNiNb_0.6_	−0.212	72.5	0.152	46.7
CrFeCoNiNb	−0.221	12.0	N/A	N/A
304SS	−0.638	12.9	N/A	N/A

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
