# Peer review of "The Effect of Nb-Content on the Microstructures and Corrosion Properties of CrFeCoNiNbx High-Entropy Alloys"

_materials, 2019, doi:10.3390/ma12223716_

Round 1

Reviewer 1 Report

The manuscript is compact and the title reflects its content. However, there are some questions:

1) There are no statistics for microhardness measurements (Fig. 4).

2) The corrosion experiment could, however, be extended. Corrosion currents determined from polarization curves are debatable - what was the immersion time before experiment? Was the EOC enough stable in sulfuric acid solution? In that solution the corrosion potentials were almost the same, although the composition of alloys varied. It would be better to use the LPR method to assess the icorr or Rp values. Authors designed the experiment in the same way as in Entropy 2018, 20, 648; doi:10.3390/e20090648. I suggest that the corrosion chapter should be expanded, e.g. use other methods/techniques to support the conclusions.

3) Fig. 5 - penultimate label in the legend - should it be CrFeCoNb?

4) Fig. 7 - Anodic branch of CrFeCoNi curve is incomplete - where is the passive region?

5) Similar question to Table 5 - the addition of Nb is not beneficial for corrosion resistance in chloride environment. The higher was the content of Nb, the higher was icorr. But for CrFeCoNi Nb there was some drop in icorr - why?

Author Response

There are no statistics for microhardness measurements (Fig. 4).

Reply: More than 5 points were averaged for each alloys, Line 64.

2) The corrosion experiment could, however, be extended. Corrosion currents determined from polarization curves are debatable - what was the immersion time before experiment? Was the EOC enough stable in sulfuric acid solution? In that solution the corrosion potentials were almost the same, although the composition of alloys varied. It would be better to use the LPR method to assess the icorr or Rp values. Authors designed the experiment in the same way as in Entropy 2018, 20, 648; doi:10.3390/e20090648. I suggest that the corrosion chapter should be expanded, e.g. use other methods/techniques to support the conclusions.

Reply: The immersion time before experiment was 900 sec, it was enough for stabilizing Eoc (Line 80). This work studied the effect of Nb-content on the microstructures, hardness and corrosion properties of CrFeCoNiNbx alloys, thus, we used the same experimental procedures. We will consider the reviewer’s suggest in our following works.

3) Fig. 5 - penultimate label in the legend - should it be CrFeCoNb?

Reply: It has been corrected.

4) Fig. 7 - Anodic branch of CrFeCoNi curve is incomplete - where is the passive region?

Reply: We made a mistake. We copied and pasted only a part of the data to plot the curve. And we corrected this mistake, shown in Figure 7.

5) Similar question to Table 5 - the addition of Nb is not beneficial for corrosion resistance in chloride environment. The higher was the content of Nb, the higher was icorr. But for CrFeCoNi Nb there was some drop in icorr - why?

Reply: The morphologies of the alloys changed with Nb-content. Maybe the ratio of fcc phase decreased with increasing Nb-content, because the major corroded phase was fcc phase. This work did not analyze the ratio of fcc and hcp phases.

Reviewer 2 Report

The manuscript deals with microstructure and corrosion performance of CrFeCoNiNb alloys. It is well structured and provides all necessary information. I recommend it for publication in Materials. Minor remarks follow.

It needs to be described how the polarization curves were measured. It should be done from the open circuit potential (Eoc) and separate measurements should be carried out for anodic and cathodic branches. If done in a single pass, explain why and discuss possible consequences, please.

The English should be improved. There are some odd formulations such as “the corrosion resistance of … slightly decreased after adding niobium, but still better than that of…” (line 14), “the FCC phase of these alloys were severely corroded than Laves phase” (line 17), “these literatures prove” (line 36), “there are two phases existed” (line 109), “almost straight increases” (line 156), “is easy to enter the passivation region than the other alloy” (line 171), “adding Nb could stable the passivation region” (line 172), “the dendrites were severely corroded than the interdendrites” (line 186), “the FCC phase was thus preferred to corroded” (line 187), “kept their origin shapes” (line 190), “the FCC phase in the CrFeCoNiNbx alloys was severely corroded than the Laves phase” (line 256), etc.

It is more common to write fcc and hcp in lowercases.

Line 64. What “kgf” refers to?

Line 69-70. “The polarization data were compared with those of commercial 304 stainless steel (304SS)…” I do not think 304 is the best reference material. All examined alloys were significantly richer in corrosion-improving elements Cr+Ni+Co and containing less iron. Thus, it is no surprise they were more stable. It would be better to select a commercial material with a composition closer to those of the experimental materials. This aspect should be mentioned and discussed in the paper.

Line 75-76. “All the potentials that are below a saturated silver chloride electrode (SSE, VSSE), whose reduction potential is 222 mV higher than that of the standard hydrogen electrode (SHE) at 25 °C.” The sentence is difficult to understand. Do the authors mean that all potentials are given to SSE, which has the redox (reduction-oxidation) potential 222 mV higher than that of SHE? Please, re-formulate.

Line 78-79. It should be explained why 1M sulphuric acid and sodium chloride solutions have been selected for the electrochemical measurement. Is it linked to any particular application of the alloys?

Line 119. Figure No. is missing.

Figure 3 is little informative. Please, think about reducing the size or, better, fully discarding it.

Line 164 and below & line 205 and below. According to ISO 8044 “Corrosion of metals and alloys - Basic terms and definitions”, you should use the term open circuit potential or free corrosion potential. Corrosion potential is ANY potential a corroding system can reach, i.e., even under polarization. Similarly, free corrosion current density is obtained at free corrosion potential.

Table 4 & 5. Not all symbols are explained in the text.

Line 252. The abbreviation HV is not explained.

Line 255. The authors claim that “adding niobium into CrFeCoNiNbx alloys could stabilize and expand the passivation regions of these alloys”. This is indeed not the case in 1 M H2SO4. It might somewhat work in 1 M NaCl, although the effect is not very pronounced. This should be made clear in the Conclusions.

Author Response

The manuscript deals with microstructure and corrosion performance of CrFeCoNiNb alloys. It is well structured and provides all necessary information. I recommend it for publication in Materials. Minor remarks follow.

It needs to be described how the polarization curves were measured. It should be done from the open circuit potential (Eoc) and separate measurements should be carried out for anodic and cathodic branches. If done in a single pass, explain why and discuss possible consequences, please.

Reply: The method of polarization test has been added into the experimental procedure, Line 81.

The English should be improved. There are some odd formulations such as “the corrosion resistance of … slightly decreased after adding niobium, but still better than that of…” (line 14), “the FCC phase of these alloys were severely corroded than Laves phase” (line 17), “almost straight increases” (line 156), “is easy to enter the passivation region than the other alloy” (line 171), “adding Nb could stable the passivation region” (line 172), “the dendrites were severely corroded than the interdendrites” (line 186), “the FCC phase was thus preferred to corroded” (line 187), “kept their origin shapes” (line 190), “the FCC phase in the CrFeCoNiNbx alloys was severely corroded than the Laves phase” (line 256), etc.

Reply: Thank you very much. We have corrected them all.

It is more common to write fcc and hcp in lowercases.

Reply: They have been changed.

Line 64. What “kgf” refers to?

Reply: The loading force for hardness test was 30 kgf (294 N), Line 64.

Line 69-70. “The polarization data were compared with those of commercial 304 stainless steel (304SS)…” I do not think 304 is the best reference material. All examined alloys were significantly richer in corrosion-improving elements Cr+Ni+Co and containing less iron. Thus, it is no surprise they were more stable. It would be better to select a commercial material with a composition closer to those of the experimental materials. This aspect should be mentioned and discussed in the paper.

Reply: We select 304 stainless steel as the reference, because it is a common commercial steel. And the results could be easily understand for the readers.

Line 75-76. “All the potentials that are below a saturated silver chloride electrode (SSE, VSSE), whose reduction potential is 222 mV higher than that of the standard hydrogen electrode (SHE) at 25 °C.” The sentence is difficult to understand. Do the authors mean that all potentials are given to SSE, which has the redox (reduction-oxidation) potential 222 mV higher than that of SHE? Please, re-formulate.

Reply: The potentials in the figures and tables were all convert to V vs SHE (standard hydrogen electrode).

Line 78-79. It should be explained why 1M sulphuric acid and sodium chloride solutions have been selected for the electrochemical measurement. Is it linked to any particular application of the alloys?

Reply: Sulfuric acid solution was selected because the standard reduction-oxidation reaction was tested in sulfate solutions; and sodium chloride solution was selected because of the ocean.

Line 119. Figure No. is missing.

Reply: It has been corrected.

Figure 3 is little informative. Please, think about reducing the size or, better, fully discarding it.

Reply: We have reduced their sizes. These TEM images are used to prove the phases of dendrites in the alloys.

Line 164 and below & line 205 and below. According to ISO 8044 “Corrosion of metals and alloys - Basic terms and definitions”, you should use the term open circuit potential or free corrosion potential. Corrosion potential is ANY potential a corroding system can reach, i.e., even under polarization. Similarly, free corrosion current density is obtained at free corrosion potential.

Reply: They have been corrected.

Table 4 & 5. Not all symbols are explained in the text.

Reply: There have been corrected, Line 170.

Line 252. The abbreviation HV is not explained.

Reply: HV means the data of Vicker’s hardness. It has been modified, Figure 4.

Line 255. The authors claim that “adding niobium into CrFeCoNiNbx alloys could stabilize and expand the passivation regions of these alloys”. This is indeed not the case in 1 M H2SO4. It might somewhat work in 1 M NaCl, although the effect is not very pronounced. This should be made clear in the Conclusions.

Reply: We mentioned it, conclusion 3.

Reviewer 3 Report

Introduction needs the description on engineering application of this alloy system. line 80; It needs the deaearation rate and time. line 107; optical microstructure line 119; check the figure number. line 153; every alloy system needs the divided line(a little confused). line 162; I think x-axis is not Nb-content. x ratio! Please clear it. Would you make the trend line and equation? line 178; Figure 5 contains the same symbol - CrFeCoNi. Please correct it. Eb is the oxygen evolution potential. Discuss it.  line 203; Figure 6 needs the photo about 304 SS. line 218; why did you finish shortly for CrFrCoNi specimen? In neutral solution, primary anodic peak should be not formed. Your peak may be occurred because you did test after the sulfuric solution test. Please check the test and correct it. CrFeCoNiNb needs the ratio of Nb. line 241; Figure 8 needs the photo of 304SS. conclusions in not the summary. please conclude your results and discussion.

Author Response

Introduction needs the description on engineering application of this alloy system.

Reply: The description has been list in conclusion 4.

line 80; It needs the deaearation rate and time.

Reply: It has been corrected, 10 sccm/min (Line 79).

line 107; optical microstructure

Reply: It has been corrected, SEM microstructures.

line 119; check the figure number.

Reply: It has been corrected.

line 153; every alloy system needs the divided line (a little confused).

Reply: It has been modified.

line 162; I think x-axis is not Nb-content. x ratio! Please clear it. Would you make the trend line and equation?

Reply: It has been corrected, Figure 4. The trend line, equation and R-square are also included in this figure.

line 178; Figure 5 contains the same symbol - CrFeCoNi. Please correct it. Eb is the oxygen evolution potential. Discuss it.

Reply: It has been corrected.

line 203; Figure 6 needs the photo about 304 SS.

Reply: This work studied the corrosion properties of CrFeCoNiNbx alloys, and the 304 stainless steel was just selected to be a reference. Let the readers easily understand the merit of CrFeCoNiNbx alloys. So we did not observe the surface of 304SS after polarization test. Also we did not show the microstructure of 304SS in Figure 1.

line 218; why did you finish shortly for CrFrCoNi specimen? In neutral solution, primary anodic peak should be not formed. Your peak may be occurred because you did test after the sulfuric solution test. Please check the test and correct it. CrFeCoNiNb needs the ratio of Nb.

Reply: We made a mistake. We copied and pasted only a part of the data to plot the curve. And we corrected this mistake, shown in Figure 7. CrFeCoNiNb is an equal molar alloy, so we do not mark the ratio.

line 241; Figure 8 needs the photo of 304SS.

conclusions in not the summary. please conclude your results and discussion.

Reply: Like the question above, this work studied the corrosion properties of CrFeCoNiNbx alloys, and the 304 stainless steel was just selected to be a reference. Let the readers easily understand the merit of CrFeCoNiNbx alloys. So we did not observe the surface of 304SS after polarization test. Also we did not show the microstructure of 304SS in Figure 1.